# Identifying the Vertical Stratification of Sediment Samples by Visible and Near-Infrared Spectroscopy

**DOI:** 10.3390/s24206610

**Published:** 2024-10-14

**Authors:** Pingping Fan, Zongchao Jia, Huimin Qiu, Hongru Wang, Yang Gao

**Affiliations:** 1Institute of Oceanographic Instrumentation, Qilu University of Technology (Shandong Academy of Sciences), Qingdao 266061, China; fanpp_sdioi@126.com (P.F.); qiuhm@qlu.edu.cn (H.Q.); wanghr@qlu.edu.cn (H.W.); 2Laoshan Laboratory, Qingdao 266237, China; 3College of Engineering, Ocean University of China, Qingdao 266404, China; zongchao_j@126.com

**Keywords:** sediment, spectroscopy, South China Sea, vertical profile, unsupervised clustering

## Abstract

Vertical stratification in marine sediment profiles indicates physical and chemical sedimentary processes and, thus, is the first step in sedimentary research and in studying their relationship with global climate change. Traditional technologies for studying vertical stratification have low efficiency; thus, new technologies are highly needed. Recently, visible and near-infrared spectroscopy (VNIR) has been explored to rapidly determine sediment parameters, such as clay content, particle size, total carbon (TC), total nitrogen (TN), and so on. Here, we explored vertical stratification in a sediment column in the South China Sea using VNIR. The sediment column was 160 cm and divided into 160 samples by 1 cm intervals. All samples were classified into three layers by depth, that is, 0–50 cm (the upper layer), 50–100 cm (the middle layer), and 100–160 cm (the bottom layer). Concentrations of TC and TN in each sample were measured by Elementa Vario EL III. Visible and near-infrared reflectance spectra of each sample were collected by Agilent Cary 5000. A global model and several classification models for vertical stratification in sediments were established by a Support Vector Machine (SVM) after the characteristic spectra were identified using Competitive Adaptive Reweighted Sampling. In the classification models, K-means clustering and Density Peak Clustering (DPC) were employed as the unsupervised clustering algorithms. The results showed that the stratification was successful by VNIR, especially when using the combination of unsupervised clustering and machine learning algorithms. The correct classification rate (CCR) was much higher in the classification models than in the global model. And the classification models had a higher CCR using K-means combined with SVM (94.8%) and using DPC combined with SVM (96.0%). The higher CCR might be derived from the chemical classification. Indeed, similar results were also found in the chemical stratification. This study provided a theoretical basis for the rapid and synchronous measurement of chemical and physical parameters in sediment profiles by VNIR.

## 1. Introduction

Sediments are one of the three most important components in marine ecosystems, recording information about the ocean’s past and future and regulating the ocean’s health. Marine sediments undergo a long and complex deposition process, forming different layers vertically [1,2,3]. These layers have distinctive physical and chemical characteristics, which are related to different carbon cycling processes and global climate change [4,5]. Therefore, vertical stratification in marine sediment profiles is the first step in sedimentary research and in studying their relationships with global climate change [6,7,8].

Traditionally, stratifying vertical profiles in marine sediments is usually carried out in a lab and is laborious, time-consuming, and costly [9,10]. Therefore, new technologies are urgently needed for rapidly determining vertical stratification [9,10]. Visible and near-infrared reflectance spectroscopy (VNIR) is a green technology with fast speeds, high sensitivity, and easy operation, being widely used in many fields [7,8]. In sediments or soils, many achievements have been made in spectral analysis of parameters by VNIR, e.g., clay content, particle size, total carbon, organic carbon, and total nitrogen [6,11,12,13,14].

Some studies have explored the use of VNIR on soil taxonomic classification or soil chronosequence classification by machine learning techniques. Using a Support Vector Machine (SVM), Zheng et al. (2019) studied soil chronosequence classification and achieved an average classification accuracy of 93.1% [6]. Jiang et al. (2021) built soil horizons using a deep learning model based on the U-net network architecture, reaching an average classification accuracy of 83% [15]. Zhang et al. (2021) developed a random forest model to identify soil horizons using a radial basis kernel SVM and achieved an accuracy of more than 70% [16]. These studies demonstrated the efficacy of VNIR in stratifying soil profiles [7,8].

Few studies have reported vertical stratification by VNIR in sediments, especially in marine sediments. Since sediments are similar to soils in terms of deposition processes and chemistry, studies on soil taxonomic/chronosequence classification could provide a sufficient reference for sediment profile stratification. Here, we explored the vertical stratification of sediments in the South China Sea by VNIR using different strategies and algorithms.

## 2. Materials and Methods

### 2.1. Sampling and Preparation

A sediment column was collected from the Northern Slope of the South China Sea (18°53.03′ N, 114°47.29′ E) in September 2020 using an SDIOI-SC100 gravity column sampler (Figure 1). This is one of the most important tropical ecosystems in China, including coral reefs, mangroves, seagrass beds, etc. It is rich in resources for the biological carbon pump [17]. The sampler was constructed from stainless steel and had a weight of 100 kg. The length and inside diameter of the inner tube of the sampler were 180 cm and 12 cm, respectively.

The sediment column was 160 cm long (Figure 2). Here, we explored rapid classification for sediment vertical profiles by spectroscopy, so we used a random physical segmentation. The column was segmented at 1 cm intervals. All samples were classified into three layers, that is, 0–50 cm (the upper layer), 50–100 cm (the middle layer), and 100–160 cm (the bottom layer). Then, all samples were freeze-dried, ground, and sieved. Each sample was divided into two parts: one for spectrum collection and the other for chemical analysis.

### 2.2. Data Collection

The visible and near-infrared reflectance spectra of these samples were collected using an Agilent Cary 5000 (Specifications see Appendix A). Cary 5000 was equipped with a special diffuse reflectance module designed for powdered samples, known as DRA-2500 (Agilent). The wavelength interval was set at 1 nm, and the scanning speed was set to 1200 nm/min. For each sample, five replicates of spectral data were collected. Consequently, the average spectra were utilized for each sample in the subsequent analyses. The concentrations of total carbon (TC) and total nitrogen (TN) in each sample were determined using a Vario EL III Elemental Analyzer at the Institute of Botany, Chinese Academy of Sciences. A total of 160 sample sets were collected. These 160 sample sets were analyzed, as shown in Figure 3.

### 2.3. Spectral Analysis

All reflectance spectra were pretreated using Savitzky–Golay (SG) filtering, with a window size of 10 wavelengths and a polynomial order of 2. Then, a global model and classification models were established to identify the stratification of each sample.

#### 2.3.1. Global Model

All samples were divided into a calibration set and a validation set, with a ratio of 2:1 by Kennard-Stone (K-S) [18]. Then, Competitive Adaptive Reweighted Sampling (CARS) was employed to identify the characteristic spectra [19]. Finally, a Support Vector Machine (SVM) and Partial Least Squares Regression (PLSR) were used to establish the spectral stratification model [20].

CARS was first proposed by Li et al. (2009) [19]. Specifically, N iterative Monte Carlo sampling was used to generate N wavelength subsets first. Secondly, the regression coefficient of each wavelength feature was calculated within each subset of the PLSR model. The regression coefficient of the wavelength represents the importance score of the feature. The regression coefficient for a wavelength serves as an indicator of the feature’s importance. Thirdly, the results were ranked in descending order based on the absolute value of these regression coefficients. Then, the root mean square error of cross-validation (RMSECV) for each wavelength subset was determined using an exponential decreasing function. After N Monte Carlo sampling was completed, the wavelength subset with the smallest RMSECV was selected as the characteristic wavelength.

An SVM is the most commonly used classification method [20]. It maps the vector into a higher-dimensional space through the kernel function to construct an optimal classification hyperplane. It identifies two parallel hyperplanes that are maximally distant and aligned with the classification hyperplane. Then, a hyperplane is defined by the equation $f\left(x\right)=\omega x+b=0$, where ω is the normal vector of the classification plane, b is the bias of the classification plane, and $f\left(x\right)=\omega x+b$ is the classification function. The greater the distance between parallel hyperplanes is, the higher the classification accuracy of the classifier is [21]. An SVM follows the principle of structural risk minimization, which can significantly reduce the occurrence of locally optimal solutions and nonlinear overfitting when dealing with a small number of samples in other machine learning algorithms [8].

PLSR is a widely used multivariate statistical analysis technique in scientific research. Compared to traditional multivariate linear regression and principal component regression, PLSR has the advantage of both effectively compressing spectral data and fully analyzing spectral data. Consequently, spectral models could be established with enhanced stability and superior prediction capabilities [22].

In this study, we selected the Gaussian radial basis function as the kernel function in SVM models. It projects the data into a higher-dimensional space, thereby effectively solving the problem of linear inseparability in the original data space. All data analyses were conducted in Matlab R2021a.

#### 2.3.2. Classification Model

Unsupervised clustering was used to divide all spectra into two subsets. These subsets were defined by K-means clustering and the Density Peak Clustering (DPC). In each subset, a spectral vertical stratification model was constructed as a global model.

K-means clustering is a distance-based clustering algorithm. Its basic idea is to find a partition scheme for K clusters interactively so the loss function corresponding to the clustering can be minimized. Here, the loss function was defined as the sum of error squares for the distance between each sample and the central point of the cluster:(1)$$J\left(c,\mu \right)=\sum_{i=1}^{M}{\left||x_{i}-\mu_{c_{i}}|\right|}^{2}$$
where $x_{i}$ is the sample i, $c_{i}$ is the cluster $x_{i}$ belongs to, $\mu_{c_{i}}$ is the central point corresponding to the cluster, and M is the total number of samples.

DPC is an unsupervised clustering algorithm based on distance and density. The selection and number of clustering centers are determined by the local density ρ and the minimum distance δ. The local density is calculated by the Gaussian kernel method [23]. The execution steps of the DPC algorithm are as follows:

(1)Calculate the distance matrix from sample set data;(2)Determine the neighborhood truncation distance $d_{c}$;(3)Calculate the local density of each point $\rho_{i},j$;



(2)
$$\rho_{i}=\sum_{i\neq j}\exp[-{\left(\frac{d_{ij}}{d_{c}}\right)}^{2}]$$


(3)
$$\delta_{i}=\underset{j:\rho_{j>\rho_{i}}}{\min}\left(d_{ij}\right)$$



(1)Calculate the offset distance of each point $\delta_{i}$;(2)Estimate the cluster center points;(3)Classify the non-clustered central data points.

#### 2.3.3. Model Evaluation

In the stratification models, the Correct Classification Rate (CCR) was used to evaluate the classification results:(4)$$CCR=\frac{TP}{N}\times 100\%$$
where TP represents the correct number, and N represents the total number. The higher the CCR is, the higher the accuracy of the classification model is [24].

Meanwhile, the determination coefficient (*R*^2^), root mean square error (RMSE), and relative predictive deviation (RPD) were also used to evaluate the spectral models.
(5)$$\begin{array}{c} R^{2}=1-\frac{\sum_{i-1}^{n}{\left(y_{i}-\hat{y_{i}}\right)}^{2}}{\sum_{i=1}^{n}{\left(y_{i}-\bar{y}\right)}^{2}} \end{array}$$
(6)$$RMSE=\sqrt{\frac{\sum_{i-1}^{n}{\left(y_{i}-\hat{y_{i}}\right)}^{2}}{n}}$$
(7)$$RPD=\frac{\sqrt{\frac{1}{n-1}\sum_{i=1}^{n}{\left(y_{i}-\bar{y}\right)}^{2}}}{\sqrt{\frac{1}{n-1}\sum_{i-1}^{n}{\left(y_{i}-\hat{y_{i}}\right)}^{2}}}$$
where $y_{i}$ represents the true value, ${\hat{y}}_{i}$ represents the modeled value, $\bar{y}$ represents the mean of the true values of all samples, and n represents the number of samples. A better model had a lower RMSE, higher *R*^2^, and larger RPD [25].

## 3. Results

We explored the use of visible and near-infrared spectroscopy in the vertical stratification of marine sediments. By employing machine learning algorithms, we established excellent spectral models to realize vertical stratification in sediment profiles in the South China Sea.

### 3.1. Physical Stratification of Sediment Profiles

All samples were classified into three layers, that is, the upper layer (0–50 cm), the middle layer (50–100 cm), and the bottom layer (100–160 cm). We first studied the spectral characteristics in the three layers and then established the stratification models.

#### 3.1.1. Spectral Characteristics in the Physical Profile

The reflectance rates of all samples were 10–50% (Figure 4a). The reflectance rates increased with wavelength at the range of 10–40% in the visible band (350–750 nm) while being concentrated at 40–50% in the near-infrared band (750–2500 nm).

The three layers had different reflectance spectra (Figure 4b). The bottom layer (100–160 cm) had the largest reflectance rates compared to the other layers. The upper layer (0–50 cm) had the smallest reflectance rates but had no significant difference compared to the middle layer (50–100 cm).

Principal component analysis (PCA) also showed the differences among these layers (Figure 5). The first two principal components could explain the difference in reflectance among the three layers, in which PC1 explained 82.3% of the difference and PC2 explained 12.0% of the difference. These results supported the evidence on the classification of the three layers.

#### 3.1.2. Physical Stratification Models

Classification models could greatly improve the performance of physical stratification, especially after the extraction of characteristic spectra. The Correct Classification Rate (CCR) was higher in the classification model than in the global model (Table 1 and Table 2). The CCR was 70.6% in the global model and 76.9–88.3% in the classification models without any characteristic spectrum extraction algorithms (Table 1). After CARS was used, the CCR increased to 76.5% in the global model and >92.6 in the classification models (Table 2).

The use of unsupervised clustering could produce a priming effect of CARS on the CCR. In the classification models, CARS played a more important role (Table 2). After using K-means combined with CARS, the CCR increased from 78% to more than 92.6%. And after using DPC combined with CARS, the CCR increased to more than 95.8%. Compared to K-means, the use of DPC combined with CARS could generate a larger CCR.

### 3.2. Physical Stratification Is Correlated with Chemistry in Sediment Profiles

The formation of physical layers in marine sediments is highly related to marine chemistry, especially organic matter. Therefore, we studied the chemical characteristics in the sediment profile, including total carbon (TC) and total nitrogen (TN).

The concentrations of TC and TN in the profile are shown in Figure 6. The distribution of TC and TN is complicated and unclear. There were no distinctive layers according to the distribution of TC or TN, respectively.

However, considering both TC and TN simultaneously, the difference among the three layers was significant. As shown in Figure 7, the upper layer had the smallest TC concentrations and the largest TN concentrations. The middle layer had the largest TC concentrations and the largest TN concentrations. The bottom layer had the largest TC concentrations and the smallest TN concentrations. Although the TC concentrations were not different among the layers, the TN concentrations were significantly different (Figure 7). These results indicated that the three layers had different chemistry.

Indeed, TC and TN could be accurately determined by spectroscopy. In our study, spectral models for TC and TN performed better for each layer (Figure 8). For the TC models, the determination coefficient (*R*^2^) was as high as 0.96, and the relative prediction deviation (RPD) was as high as 4.7 (Figure 8c). The TN models performed as well as the TC models (Figure 8).

In addition, we further cross-validated the classification spectral models across all layers. For example, we used the TC model of the upper layer to predict the samples in the middle layer and bottom layer. The results showed that a spectral model in a specific layer was not well suited for samples in other layers (Table 3). These results also support the results of physical stratification.

## 4. Discussion

Our study showed that the marine sediment vertical profile exhibited a complicated but regular structure and morphology. Visible and near-infrared spectroscopy (VNIR) is a green technology with fast speeds, high sensitivity, and easy operation. And VNIR has been proven to be an efficient technology in many fields. There are some studies on soil taxonomic classification or soil chronosequence that have used VNIR [6,7,8,15,16], but application on sediment vertical stratification is rare. Our study showed that the physical layer could be classified by spectroscopy, mainly because these layers were highly related to chemical properties, such as TC and TN. Our study provided a new application for visible and near-infrared spectroscopy (VNIR) on marine sediments. In the future, more parameters should be developed by VNIR synchronously, which could make marine sediments easily “seen” in the sea.

In our study, stratification models performed better than the past studies on soil profiles. In our study, the *R*^2^ of spectral models could reach as high as 0.96, and the correct classification rate (CCR) achieved up to 96%. In past studies on soil profiles, the highest CCR was 93.1% [6], which is smaller than that in our study and may be because effective algorithms were used in our study.

Obviously, algorithms play an important role in establishing excellent spectral models. Twenty years ago, Confalonieri et al. (2001) and Smith et al. (2001) compared the influence of different algorithms on spectral models [26,27]. Mouazen et al. (2010) found that machine learning algorithms had the advantage of establishing spectral models for soil nutrients [28]. Jia et al. (2014) proposed that the predominant step in establishing spectral models for soil nutrients was extracting the characteristic spectra [29]. In our study, CARS was used as the characteristic spectra extraction algorithm and showed a great effect on spectral models. And the unsupervised clustering of DCP performed better than K-means in classification models. These studies revealed the importance of algorithms in establishing spectral models.

This study showed that the combination of unsupervised clustering and CARS effectively improved the performance of vertical stratification. Because these algorithms played different roles in spectral analysis, their combination could nonlinearly amplify their different roles. Therefore, CARS showed a priming effect on vertical stratification after using unsupervised clustering in our study.

The results also highlighted the effectiveness of combining unsupervised clustering with machine learning to construct stratification models for sediments or soils. This was also suggested in other studies [6,30]. For example, Jiang et al. (2021) built soil horizons using a deep learning model based on the U-net network architecture, reaching an average classification accuracy of 83% [15].

Our study explored a new application of VNIR on marine sediments. Here, we used random physical segmentation to explore the rapid classification of sediment vertical profiles. In the future, we will increase the measurement accuracy of both carbon and nitrogen to improve the rapid determination of other related parameters by VINR, for example, how to scientifically segment a sediment profile and calculate the sedimentation rates using VNIR.

## 5. Conclusions

We explored vertical stratification in a sediment column sampled from the South China Sea by establishing a global model and classification models. Both the global model and classification models were established by a Support Vector Machine (SVM) and Partial Least Squares Regression (PLSR) after the characteristic spectra were identified using Competitive Adaptive Reweighted Sampling (CARS). In the classification models, K-means clustering and Density Peak Clustering (DPC) were employed as the unsupervised clustering algorithms. The results showed that physical stratification had a correct classification rate (CCR) of 76.5% in the global model. In the classification models, the CCR was 94.8% using K-means combined with CARS and 96.0% using DPC combined with CARS. Physical stratification could be influenced by chemical properties, such as total carbon and total nitrogen. These results suggested that vertical stratification in marine sediments could be rapidly identified by VNIR, especially when using the combination of unsupervised clustering and machine learning algorithms. This study provided an important technological basis for simulating marine sedimentation processes and establishing marine C cycling models.

## Figures and Tables

**Figure 1 sensors-24-06610-f001:**
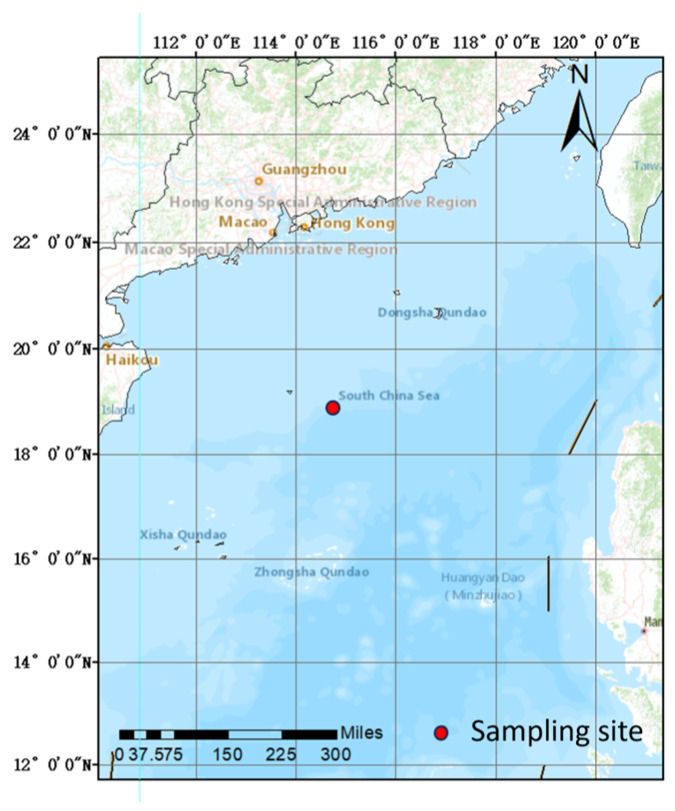
Sampling site in the South China Sea.

**Figure 2 sensors-24-06610-f002:**
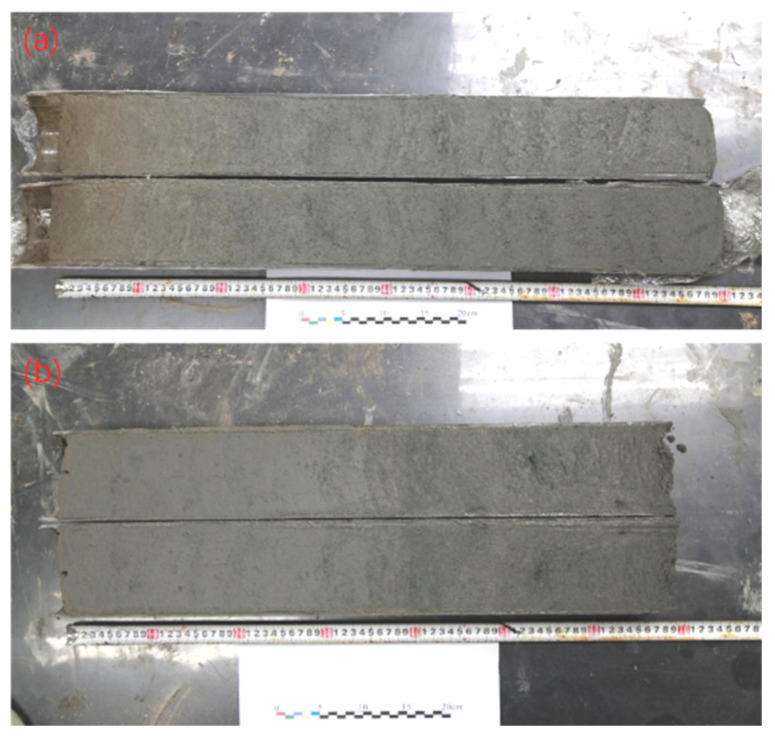
Sediment column. (**a**) 0–80 cm. (**b**) 81–160 cm.

**Figure 3 sensors-24-06610-f003:**
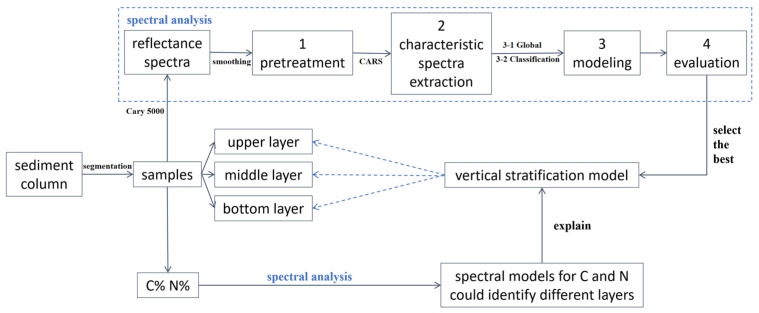
A flow diagram of data analysis in this study.

**Figure 4 sensors-24-06610-f004:**
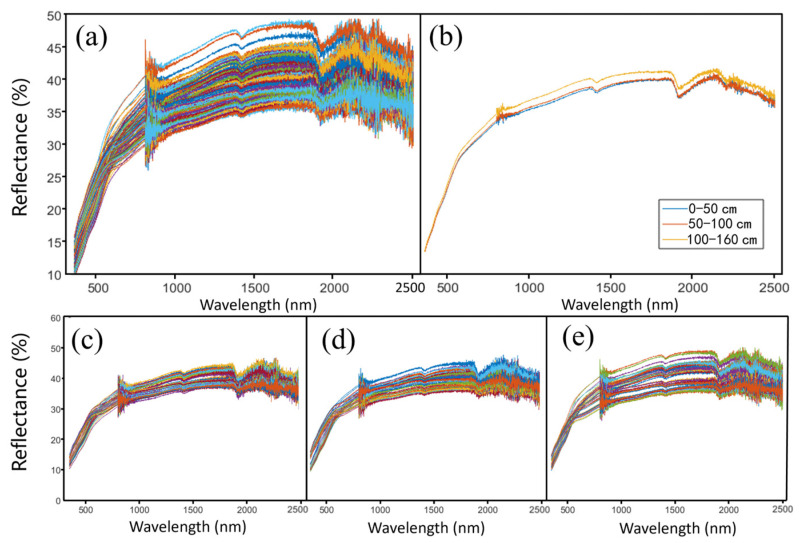
Reflectance spectra of sediments in all samples and different layers. (**a**) All samples. (**b**) The average spectrum of each layer. (**c**) The upper layer. (**d**) The middle layer. (**e**) The bottom layer.

**Figure 5 sensors-24-06610-f005:**
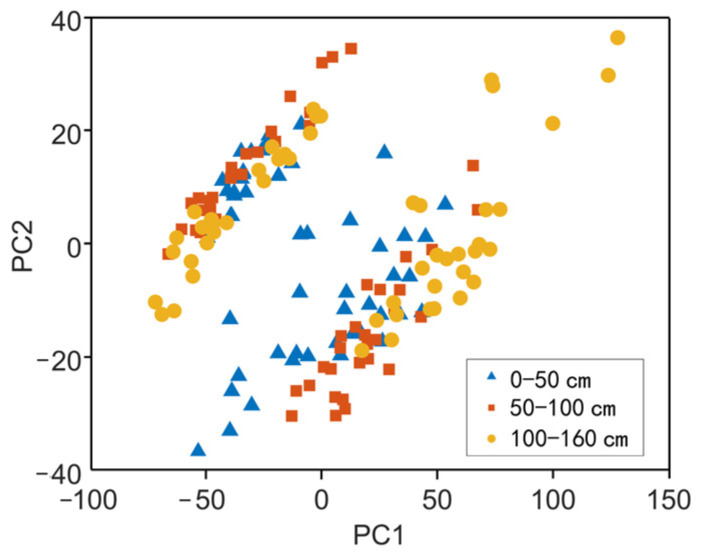
Principle component analysis of reflectance spectra in sediments of different layers.

**Figure 6 sensors-24-06610-f006:**
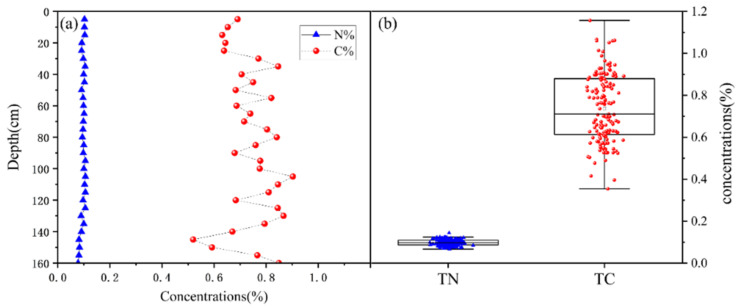
Chemical distribution of total carbon and total nitrogen. (**a**) Concentrations of total carbon and total nitrogen in the sediment profile, (**b**) boxplot of total carbon and total nitrogen.

**Figure 7 sensors-24-06610-f007:**
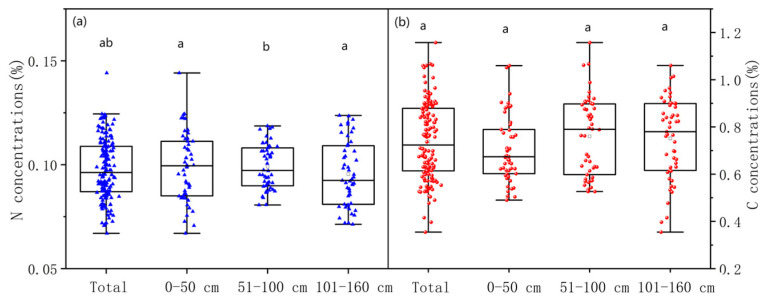
Boxplot of concentrations of total nitrogen (**a**) and total carbon (**b**) in all samples and different layers. In each subfigures, the characters “a”, “b”, and “ab” are the results of ANOVA for average concentrations of total carbon and total nitrogen.

**Figure 8 sensors-24-06610-f008:**
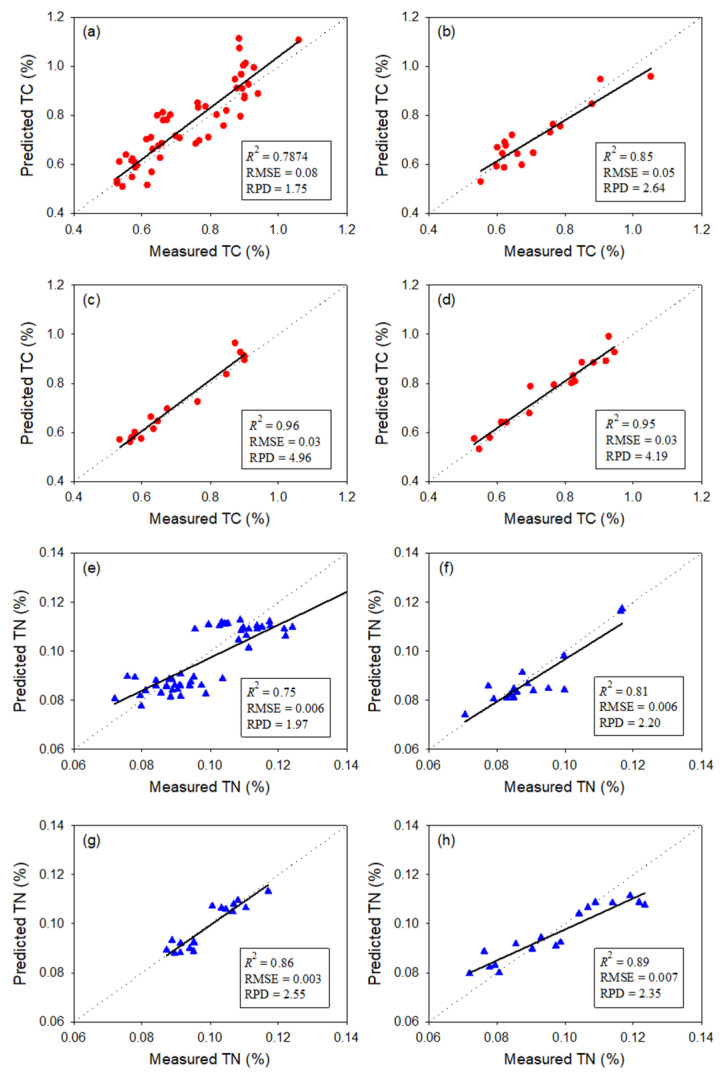
Results of total carbon and total nitrogen by global model and classification models. (**a**) TC in all samples, (**b**) TC in the upper layer, (**c**) TC in the middle layer, (**d**) TC in the bottom layer, (**e**) TN in all samples, (**f**) TN in the upper layer, (**g**) TN in the middle layer, (**h**) TC in the bottom layer. In subfigures, the red circles and blue triangles were TC data and TN data, respectively; in the dotted lines (1:1 lines) the predicted values equals to the measured values; the solid lines were the fitted curves whose parameters were listed in the left box.

**Table 1 sensors-24-06610-t001:** Correct classification rates (CCRs) in the global model.

Characteristic Spectra Selection Method	Correct Classification Rates (CCRs)	
	Calibration Set	Validation Set
None	100%	70.6%
Competitive Adaptive Reweighted Sampling	100%	76.5%

**Table 2 sensors-24-06610-t002:** Correct classification rates (CCRs) in the classification models.

Clustering Method	Subset	Characteristic Spectra Selection Method	Correct Classification Rates (CCRs)	
			Calibration Set	Validation Set
K-means	1	None	100%	77.8%
		Competitive Adaptive Reweighted Sampling	100%	92.6%
	2	None	100%	78.3%
		Competitive Adaptive Reweighted Sampling	97.9%	95.7%
Density Peak Clustering	1	None	100%	76.9%
		Competitive Adaptive Reweighted Sampling	100%	96.2%
	2	None	100%	83.3%
		Competitive Adaptive Reweighted Sampling	100%	95.8%

**Table 3 sensors-24-06610-t003:** Validation results of TC and TN in sediments of all three layers from the spectral model established using the samples in a specific layer.

Spectral Model	Sample	TC			TN		
		*R* ^2^	RMSE	RPD	*R* ^2^	RMSE	RPD
Upper layer	Upper layer	0.85	0.05	2.64	0.81	0.006	2.20
	Middle layer	0.26	0.17	0.96	0.63	0.008	1.22
	Bottom layer	0.15	0.21	0.81	0.59	0.013	1.28
Middle layer	Upper layer	0.22	0.16	0.85	0.39	0.014	1.24
	Middle layer	0.96	0.03	4.66	0.86	0.003	2.55
	Bottom layer	0.35	0.15	1.13	0.66	0.010	1.23
Bottom layer	Upper layer	0.35	0.15	1.13	0.33	0.014	1.23
	Middle layer	0.43	0.13	1.31	0.38	0.008	1.19
	Bottom layer	0.95	0.03	4.19	0.89	0.007	2.35

## Data Availability

Data are contained within the article.

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
