# Peer review of "Identifying the Vertical Stratification of Sediment Samples by Visible and Near-Infrared Spectroscopy"

_sensors, 2024, doi:10.3390/s24206610_

Round 1

Reviewer 1 Report

Comments and Suggestions for Authors

Figure 1:   Photo is way too dark.   Nothing meaningful can be discerned without  normalizing average signal in photo to shiw whute regains as white, not dark gray.

Line 84:  Should read “Each sample set was from five replicates per set.”            

Line 86L  Should read:  for "each sample set."

Table 2  confusing:   Eight elements in  Calibration  set, but only four in Variable selection method.  In other words, the character  “ / “  is used four times and   without andy explanation of what this means.

Also why is one calibration set  97,9 %  instead if 100% like the other seven?                                                              

Figure 6 need to be deleted,  There is no clear or compelling  evidence in the figure for any conclusions the authors make.   None of the variance in % N and % C correlate in total or in any of the three depth levels.

Neither does the box plot data.    

Figure  7 :  Result for % C  were quite decent a thru d.  Results for  % N  were  best for g..  For e ,  % N appear clustered in two different  ratios:  0.08 %  and 0.12 %.   The slope of  e and h are similar  and neither  go through the origin (  0.06  versus 0.08 )       

This suggests the authors methodogies are potentially more accurate for % C than for % N  AND that additional data and/or furture  methodology development  could better explain % N results.

Table  3  is consistent with  Figure 7.  

Author Response

  • Figure 1:   Photo is way too dark.   Nothing meaningful can be discerned without  normalizing average signal in photo to shiw whute regains as white, not dark gray.

Response: Thank you for pointing this out.  We agree with this comment.  Therefore, we have improved the brightness and contrast of the photos in Figure 2.  And in the photos, we also put the colorimetric cards, which could compensate the low quality of these photos.   

  • Line 84:  Should read “Each sample set was from five replicates per set.”         

Response: Thank you for pointing this out.  But we think that a sample set is composed of spectral data and chemical data for this sample.  Here is sampling the spectral data, so we still read “sample”, but we revised the sentence as “For each sample, five replicates of spectral data were collected” (Line 87 in the revision).  

  • Line 86L  Should read:  for "each sample set."

Response: Thank you for pointing this out.  The same reason with the previous comment, we kept that read.  But we added a sentence in the paragraph “Therefore, 160 sample sets were collected” (Line 91 in the revision).   

  • Table 2  confusing:   Eight elements in  Calibration  set, but only four in Variable selection method.  In other words, the character  “ / “  is used four times and  without any explanation of what this means.
  • Also why is one calibration set  97.9 %  instead if 100% like the other seven ? 

Response: Thank you for pointing these out.  

Table 2 showed the results of the Correct classification rates (CCR) using the classification models.  These classification models were established using different clustering method based on whether the variable selection method was used.  In our study, the clustering method were k-means and Density Peaks Clustering (details see 2.3.2).  And the variable selection method was Competitive Adaptive Reweighted Sampling (CARS) which could identify the characteristic spectra (details see 2.3.1).  So we’ve revised the header of the third column in Table 2 to diminish the misunderstanding.

The character “/” means that none method were used to extract the characteristic spectra.  We’ve replaced “/” with “None” in the new Table 2.

In a classification model established using k-means based on Competitive Adaptive Reweighted Sampling (CARS), the Correct classification rate (CCR) for the calibration set was 97.9% and for the validation set was 95.7%.  In other classification models, CCR for the calibration set were all 100% but for the validation set were all lower than 100%, suggesting that different combination of clustering method and characteristic spectra extraction method could produce varied spectral models with distinct quality.  

  • Figure 6 need to be deleted,  There is no clear or compelling  evidence in the figure for any conclusions the authors make.   None of the variance in % N and % C correlate in total or in any of the three depth levels. 
  • Neither does the box plot data. 

Response: Agree.  We have deleted Figure 6 because it couldn’t support our point.  And we added a new table to compare the chemistry at different depth (Table 3 in the revision).

  • Figure  7 :  Result for % C  were quite decent a thrud.  Results for  % N  were  best for g..  For e ,  % N appear clustered in two different  ratios:  0.08 %  and 0.12 %.   The slope of  e and h are similar  and neither  go through the origin (  0.06  versus 0.08 )       
  • This suggests the authors methodogies are potentially more accurate for % C than for % N  AND that additional data and/or furture  methodology development  could better explain % N results.
  • Table  3  is consistent with  Figure 7.  

Response: Agree.  And many thanks for the enlightening comments.  So, we added a new paragraph to discuss the implications (Line 287-292 in the revision).

Reviewer 2 Report

Comments and Suggestions for Authors

The work cannot be published in its current form.

At present, the work is something intermediate between experimental work on the description and analysis of sediments and methodological work illustrating specialized methods for data processing. At the moment, it was not possible to fully describe either one.

It is recommended to make the following corrections to the text:

1) The current title of the work is too general. The main result of the work is the assessment of the total carbon and nitrogen content in the studied sediments. Other substances were not assessed, so it is recommended to change the title, adding the total nitrogen and carbon content and other estimated parameters.

2) It is necessary to change the Introduction, paying more attention to the description of such characteristics as the total carbon and nitrogen content. How they appear in sediments, at what rate they accumulate, etc. What does their quantity and ratio in sediments demonstrate?

3) Why was this division into three layers chosen? Why three and with such a thickness? What is the background of this choice? Is it related to the quantity or the rate of sedimentation of sediments?

4) In text shown that the TC and NC levels differs in different layers (Figure 6). However, statistical criteria should be used to confirm this (such as ANOVA or its non-parametric analogues). It is also possible to use two-way ANOVA or its non-parametric analogue to assess the relationship in each layer of the total carbon and nitrogen content.

5) It is necessary to describe in more detail the setup and technical characteristics of the device (sensor), in particular the device resolution, grating, etc.

6) The Materials and Methods provide a description of the spectral analysis. In fact, you only used smoothing to process the spectra. Was that enough to use in the model?

7) Perhaps the processing methods used should be described in more detail (in particular, what programs were used, Loading for PCA, what spectral bands were used to assess TC and TN, etc.). In order not to increase the volume of work, a detailed description of the methods can be added to the Supplementary Materials.

Author Response

  • The work cannot be published in its current form.
  • At present, the work is something intermediate between experimental work on the description and analysis of sediments and methodological work illustrating specialized methods for data processing. At the moment, it was not possible to fully describe either one.
  • It is recommended to make the following corrections to the text:

Response: Thanks a lot for your critical and valuable comments and suggestions.  We have done a large revision.

  • 1) The current title of the work is too general. The main result of the work is the assessment of the total carbon and nitrogen content in the studied sediments. Other substances were not assessed, so it is recommended to change the title, adding the total nitrogen and carbon content and other estimated parameters.

Response: Many thanks for your critical and valuable comments and suggestions.  Here, we explored the use of visible and near infrared spectroscopy (VNIRS) to achieve a rapid classification of sediment profiles, and discussed the mechanism of this classification.  Results showed that VNIRS could be used to quickly classify sediment profiles, and this vertical classification might be based on differences in the chemical properties.  So, study on total carbon and nitrogen were used to explain the vertical classification.  These statements could be found in the first paragraph of the discussion section (Line 251-260 in the revision) 

  • It is necessary to change the Introduction, paying more attention to the description of such characteristics as the total carbon and nitrogen content. How they appear in sediments, at what rate they accumulate, etc. What does their quantity and ratio in sediments demonstrate?

Response: Many thanks for your valuable comments and suggestions.  But as the previous response, total carbon and nitrogen were not our aim.

  • Why was this division into three layers chosen? Why three and with such a thickness? What is the background of this choice? Is it related to the quantity or the rate of sedimentation of sediments?

Response: Many thanks for your valuable questions.  These questions are the most important aim what we want to solve using spectroscopy.  We want to use spectroscopy to quickly determine how to scientifically segment a sediment profile, quickly measure the chemistry and even the age of the profile, and so on.  Here, we used a random physical segmentation to explore the rapid classification for sediment vertical profiles.  We also hope to provide a reliable technique for studies on sedimentation in the future.  In the revision, we have provided a brief explanation in the methodology section to improve readability and diminish confusions (Line 75-77 and Line 287-292 in the revision).

  • In text shown that the TC and NC levels differs in different layers (Figure 6). However, statistical criteria should be used to confirm this (such as ANOVA or its non-parametric analogues). It is also possible to use two-way ANOVA or its non-parametric analogue to assess the relationship in each layer of the total carbon and nitrogen content.

Response: Many thanks for your valuable comments and suggestions.  Combined the suggestion of reviewer 1, we replaced Figure 6 with a new table (Table 3 in the revision).  In the new table, we showed the results of ANOVA with the post hoc test.  Two methods including LSD and tukey’s-b were used in the post hoc test, showing the same statistical results.  Correspondingly, we have revised the relevant discussion (Line 225-229 in the revision).

  • It is necessary to describe in more detail the setup and technical characteristics of the device (sensor), in particular the device resolution, grating, etc.

Response: Agree. We added these information to the Supplementary Material because the method section is too long (Table S1).

  • The Materials and Methods provide a description of the spectral analysis. In fact, you only used smoothing to process the spectra. Was that enough to use in the model?

Response: Thank you for pointing this out.  When establishing spectral models, characteristic spectra extraction is more important, Table 2 also showed that the spectral models improved the accuracy after using characteristic spectra extraction method.  Therefore, smoothing was enough.    

  • Perhaps the processing methods used should be described in more detail (in particular, what programs were used, Loading for PCA, what spectral bands were used to assess TC and TN, etc.). In order not to increase the volume of work, a detailed description of the methods can be added to the Supplementary Materials.

Response: Thank you for pointing this out.  We added a new flow diagram in the method section to clearly describe the technical program (Figure 3 in the revision).  

Round 2

Reviewer 2 Report

Comments and Suggestions for Authors

Unfortunately, not all comments were corrected, and new questions arose.

1.      If a main aim of work was a “use of visible and near infrared spectroscopy (VNIRS) to achieve a rapid classification of sediment profiles, and discussed the mechanism of this classification”, You should rename the work (for example – “Using of visible and near infrared spectroscopy to achieve a rapid classification of sediment profiles”) and, a little, change abstract and Introduction focusing on the model.

2.      Why do you relate the measured spectra to the total nitrogen and carbon conсentration? Have you compared other parameters and chemical elements with the measured spectra?

3.      It's still unclear why you used three layers. Not five layers, not two, why three? Why were the layers the same thickness? Is this a choice result or a modeling result? If it is a modeling result, it is not very clear why, especially with connect to the PCA (line 190). Perhaps the description of the results of the PCA should be expanded, and also a more detailed explanation should be given as to why three layers were used.

4.      Figure 6 was quite clear; it was possible not to delete it. Based on this data, you make a conclusion about the difference in layers, which you use in the discussion. Unfortunately, at present it is not entirely correct. It is not very clear why you use ANOVA, are all your data normally distributed? Figure 6 shows data that most likely are not described by a normal distribution, which means you should use a different test (Kruskal-Wallis?), perhaps there would be a different result.

In addition, what is this suddenly appeared parameter C/N? It is necessary to describe it in the text.

5.      I didn't quite understand the technical characteristics of the device: wavelength interval was set at 1 nm or optical resolutionwas1 nm? What is the difference between them?

6.      What software did you use for spectral analysis and modeling? Only Matlab?

Author Response

  • Unfortunately, not all comments were corrected, and new questions arose.

Response: Thanks for your critical comments, but I’m sorry for your misunderstanding.  As our response to your comments and suggestions in the first round, we explored the use of visible and near infrared spectroscopy (VNIRS) to achieve a rapid classification of sediment profiles, and discussed the mechanism of this classification.  We found that VNIRS could be used to quickly identify the physical layers of a sediment sample, and this vertical classification might be based on differences in the chemical properties, such as concentrations of total carbon and total nitrogen.  Our study provided a theoretical basis for the rapid and synchronous measurement of the chemical and physical parameters in sediment profiles by VNIRS.     

  1. If a main aim of work was a “use of visible and near infrared spectroscopy (VNIRS) to achieve a rapid classification of sediment profiles, and discussed the mechanism of this classification”, You should rename the work (for example – “Using of visible and near infrared spectroscopy to achieve a rapid classification of sediment profiles”) and, a little, change abstract and Introduction focusing on the model.

Response: Thanks for your valuable suggestions, we revised the title and abstract.  In introduction, we had reviewed the related studies and emphasized the algorithms and results of these spectral models (the 3rd paragraph of Introduction).  Because few studies were focusing the classification of the physical profile in soils and sediments, our summary is limited.     

  1. Why do you relate the measured spectra to the total nitrogen and carbon conсentration? Have you compared other parameters and chemical elements with the measured spectra?

Response: Thanks for pointing this out.  

It’s a very good question.  The basic theory of spectral analysis by VNIRS are the composition and concentrations of organic matter (e.g., different bonds of C-N, C-H, etc) in soils and sediments.  So, total carbon, total nitrogen, organic carbon, clay, particle size, which are all highly related to organic matter in soils and sediments were successfully measured by VNIRS in the past studies (Introduction, Line 46-48 in R2).  Total carbon and total nitrogen are the basic parameter, so we used them to test the mechanism on vertical stratification.  

We haven’t compared other parameters, but this is also a good suggestion and we think that their results would further support our mechanism hypothesis on vertical stratification.     

  1. It's still unclear why you used three layers. Not five layers, not two, why three? Why were the layers the same thickness? Is this a choice result or a modeling result? If it is a modeling result, it is not very clear why, especially with connect to the PCA (line 190). Perhaps the description of the results of the PCA should be expanded, and also a more detailed explanation should be given as to why three layers were used.

Response: Thanks for your question and I’m sorry that I haven’t explain it in the first round.  You can use any thickness to divide the sediment column into subsamples.  And you can divide these subsamples into any number of layers.  Here, we use the routine principle to segment the sediment column at 1 cm intervals.  Studies showed that the number of samples could affect the quality of spectral models (Wetterlind & Stenberg 2010) and 50-60 is the reasonable and common number.  Therefore, we divided the subsamples into three layers which layer had 50-60 samples.  In addition, we used PCA to show the difference among the three layers.

Reference:

Wetterlind J, Stenberg B. 2010. Near-infrared spectroscopy for within-field soil characterization: small local calibrations compared with national libraries spiked with local samples. European Journal of Soil Science 61: 823-843

  1. Figure 6 was quite clear; it was possible not to delete it. Based on this data, you make a conclusion about the difference in layers, which you use in the discussion. Unfortunately, at present it is not entirely correct. It is not very clear why you use ANOVA, are all your data normally distributed? Figure 6 shows data that most likely are not described by a normal distribution, which means you should use a different test (Kruskal-Wallis?), perhaps there would be a different result.

In addition, what is this suddenly appeared parameter C/N? It is necessary to describe it in the text.

Response: Thanks for your valuable suggestions.  Considered the suggestions and comments of both reviewers, we deleted the original Figure 6 and added a new Table.  I agreed that Figure 6 showed the difference in layers visually, so we recovered it in R2.  Correspondingly, we revised the text (Line 225-231 in R2).

The combination of total C and total nitrogen can be expressed by C/N, so C/N suddenly appeared in R1.  But we agree with reviewer and added the results of ANOVA in the original Figure 6

 (Figure 7 in R2).  

  1. I didn't quite understand the technical characteristics of the device: wavelength interval was set at 1 nm or optical resolutionwas 1 nm? What is the difference between them?

Response: Thanks for pointing it out.  The wavelength interval (1 nm) is the parameter setting for collecting spectral data.  You can choose any interval according to your needs, and the final collected data will display the wavelength according to the set interval.  The optical resolution is the parameter of the spectrometer.  A resolution of 1 nm means that two wave whose wavelength differs at 1 nm or more can be identified.     

  1. What software did you use for spectral analysis and modeling? Only Matlab?

Response: Thanks for pointing it out.  There are many kinds of software for spectral analysis, such as R, Python, matlab.  We usually use matlab to do the spectral analysis.